# Do Aging and Low Fertility Reduce Carbon Emissions in Korea? Evidence from IPAT Augmented EKC Analysis

**DOI:** 10.3390/ijerph17082972

**Published:** 2020-04-24

**Authors:** Jaehyeok Kim, Hyungwoo Lim, Ha-Hyun Jo

**Affiliations:** 1National Assembly Budget Office, 1 Uisadang-daero, Yeongdeungpo-gu, Seoul 07233, Korea; kjh9926@assembly.go.kr; 2Department of Economics, Yonsei University, 50 Yonsei-ro, Seodaemun-gu, Seoul 03722, Korea; hyungwoo.lim0206@yonsei.ac.kr

**Keywords:** carbon dioxide, age structure, population aging, environmental kuznets curve, IPAT, panel cointegration regression, fully modified ordinary least squares (FMOLS)

## Abstract

The purpose of this article is to empirically find the Environmental Kuznets Curve (EKC) relationship between income and carbon dioxide (CO_2_) emissions and to analyze the influence of population aging on such emissions. We utilize Korean regional panel data of 16 provinces during the period from 1998 to 2016. To account for the nonstationary time series in the panel, we employ a fully modified ordinary least squares (FMOLS) and estimate long-run elasticity. From the empirical results, we can find the nonlinear relationship between income and CO_2_ emissions. Additionally, we verify the fact that population aging reduces CO_2_ emissions. A 1% increase in the proportion of the elderly results in a 0.4% decrease in CO_2_ emissions. On the other hand, the younger population increases CO_2_ emissions. These results were in line with those of additional analysis on residential and transportation CO_2_ emissions, for the robustness check.

## 1. Introduction

One of the biggest goals faced by human society is that of sustainable development. During the era of rapid economic growth over the past decades, less attention was paid to environmental problems and the depletion of energy resources because economic growth took priority over those issues. However, currently, the rapid growth of the world economy requires tremendous amounts of energy sources. Until now, fossil fuels have been the major source of energy. The biggest problem with using fossil fuels is that the byproduct of their use is greenhouse gases (GHGs). While GHGs are essential for maintaining the earth’s atmospheric temperature, excess GHGs are harmful to nature. The Intergovernmental Panel on Climate Change (IPCC) has identified GHGs from fossil fuel usage as the main driver of climate change.

To explore the global climate change issue, a large amount of research has focused on examining the environment–income relationship. However, since the seminal paper [1], Environmental Kuznets Curve (EKC) hypothesis has become the standard approach to investigate the issue of sustainable development. EKC explains the relationship between economic growth and environmental degradation through an inverted U-shaped curve.

According to EKC, the environment continuously degrades as income grows until a certain stage of economic development. This is generally the case in early stages of economic development. However, beyond a threshold income level, environmental quality improves in line with growing income. Theoretical explanations for this phenomenon include the development of pollution abatement technology, changes in industrial structure from manufacturing to services, and the increased demand for a clean environment that accompanies income growth. Nonetheless, the inverted U-shaped EKC curve is not a standard form of a pollution–income relationship across all empirical studies. It has many different forms depending on the country, region, empirical methodology, and sample range.

According to existing literature on EKC, besides income, energy-related variables such as energy price and energy efficiency are considered to be important factors. Additionally, the IPAT model (method of decomposing environmental impact (I) into socio-economic variables: population (P), affluence (A), and technology (T)) suggests that socio-demographic variables such as population, and urbanization are also important drivers of environmental quality. However, among socio-demographic factors, researchers pay relatively less attention to population aging. 

In fact, population aging can affect national GHG levels in two ways. First, the behavior of older people differs from that of other age groups. They tend to stay home longer and demand more heating and cooling to maintain their health. This could increase residential energy consumption and may lead to increased emissions [2]. On the other hand, although older people spend significant time at home, they are relatively less active than other age groups and the size of aged households is small. Thus, they use less energy appliances and transportation, which could decrease energy consumption [3,4,5,6].

In either case, population aging can affect carbon emissions. Second, old people have stronger preferences for better air quality [7,8]. As life spans increase, people are more concerned about environmental quality since it directly affects their health. This affects voting patterns and eventually, support for relevant policies. 

Against this background, we examine the role of population aging on CO_2_ emissions in Korea. Similar to other developed countries, Korea has been facing the issue of population aging. Statistics Korea, the Korean statistics authority, reports that 13.2% of the total population was aged 65 and above in 2016, which is almost double of its value of 7.2% in 2000.

In addition, according to the World Development Indicator (WDI), Korea was ranked 9th in terms of total CO_2_ emissions in 2016. Korea declared a target to reduce its CO_2_ emissions by 37% from its business-as-usual (BAU) levels by 2030. Thus, investigating the effect of population aging on CO_2_ emissions is a very important policy in terms of decreasing those emissions.

The contribution of this study is as follows. First, we include age structure variables as potential drivers of CO_2_ emissions. Although the population age structure can have a direct or indirect effect on CO_2_ emissions, not many studies have taken it into consideration. Second, we investigate the environment–demographic relationship in Korea on a regional level by estimating regional CO_2_ emissions using fossil fuel consumption data. To our knowledge, this is the first paper to examine regional CO_2_ emissions in Korea. Third, we analyze the CO_2_–income relationship using the third-order polynomial of the per capita income variable, which allows us to investigate the possibility of an N-shaped curve. Fourth, in order to maintain the robustness of our empirical analysis, we further analyze the impact of age structure on CO_2_ emissions from residential and transportation sectors.

This paper is organized as follows. Section 2 reviews past literature, while Section 3 introduces the methods for estimating regional CO_2_ emissions and the IPAT model. The regression models and empirical results are shared in Section 4. Finally, Section 5 and Section 6 conclude the study.

## 2. Literature Review

Since the seminal work [1], most empirical works focus on estimating the inverted U-shaped curve. On the other hand, relatively few attempts have been made to identify an N-shaped curve. The general form of EKC hypothesis can be formulated as follows:(1)$$E_{it}=\alpha_{1}+\beta_{1}Y_{it}+\beta_{2}Y_{it}^{2}+\beta_{3}Y_{it}^{3}+\varepsilon_{it}.$$
where $E_{it}$ and $Y_{it}$ denotes indicator of environmental degradation and income level for country (or region) *i* at time *t*, respectively. Moreover, $\varepsilon_{it}$ is the error term.

### 2.1. EKC Studies

Among early efforts to establish the N-shaped curve, [9,10] conducted the EKC study using country-level panel data. In [10] investigated the pollution–income relationship for 30 countries over the period 1982–1994 by using Sulphur dioxide (SO_2_) emissions, while [9] set up a panel regression model for 16 countries for the period of 1950–1992 using CO_2_ emissions. Both find the N-shaped curve but [9] attribute the result to the data aggregation of USA, Canada, and Luxembourg. In [11] examined the N-curve relationship between income and environment quality. They employed the cointegration model using Turkish time series data from 1968–2003 and the panel data model from 1992–2001, including observations from 58 Turkish provinces. The study fails to establish an N–curve relationship based on the time series analysis for CO_2_ but the panel data model indicates an N–shape relationship for SO_2_ and PM10 (particulate matter 10 micrometers or smaller) emissions.

Some EKC studies found an N-shaped curve relationship between income and emissions [12,13,14]. In [12] considered the Austrian case and employed a time series methodology for the EKC study using CO_2_ emissions data from 1960 to 1999. Given the small, open, and industrialized nature of the economy and the industrial structural changes, they focused on import and services sector variables and discovered an N-shaped curve for the Austrian economy. In [13] applied pooled mean group estimators for 22 OECD (Organization for Economic Cooperation and Development) economies to find the N-shaped curve based on CO_2_ data from 1975 to 1998. In [14] estimated various EKC models using panel data for 109 countries from 1959 to 2001. Using the large country panel data, they applied the hierarchical Bayes estimator, which allowed for heterogeneity. In their study, they established the N-shaped curve for OECD and EU sub-samples, but not for the overall sample.

EKC studies for Korea are rare. In [15] employed the panel regression using Korean regional data for the period of 1990–2005. They chose SO_2_, CO, and nitrogen dioxide (NO_2_) emissions as the environmental degradation variable and set up the cubic regression equation. They concluded that the U-shaped curve was appropriate for CO and NO_2_, but failed to find the typical inverted U-shaped or N-shaped curves for SO_2_. In [16] employed the time-series error-correction model to investigate CO_2_ emissions. Using data spanning the period of 1971–2007, they found the inverted U-shaped curve.

### 2.2. EKC Studies with Demographic Structure

Many studies extend the EKC model by including socio-demographic variables. These studies are based on the extension of the IPAT model, which is the general method of decomposing environmental impact (I) into socio-economic variables: population (P), affluence (A), and technology (T). However, among the extensive literature on IPAT augmented EKC models, population aging is rarely considered.

The empirical results were mixed in past literature. The main results of previous studies are summarized in Table 1. Some papers argued that population aging leads to a reduction in pollution emissions [4,5,17]. In [17] set up the computable general equilibrium (CGE) type Population-Environment-Technology (PET) model to analyze how population aging influences total CO_2_ emissions from the use of fossil fuels in the US economy. They incorporated population age structure into an energy-economic model with heterogeneous households. In their analysis, aging implied a reduction in aggregate labor supply, thereby reducing long-term emissions. However, this study actually focuses on the effect of population aging on total economic activity and not on the behavior of elderly people. In [5] focused their study on 17 developed countries spanning the period of 1960–2005. They disaggregated population into four age groups: 20–34, 35–49, 50–64, and 65–79 years old. By doing so, they found that older age groups reduce CO_2_ emissions. In [4] investigated the role of age structure on carbon emissions from transportation based on data from 22 OECD countries spanning the period of 1960–2007. The study employed the panel fully modified ordinary least squares (FMOLS) to estimate the long-run impact of age structure. These results reveal that young adults (aged 20–34) have environmentally intensive effects but other age cohorts (aged 35–49, 50–69, 70+) decrease CO_2_ emissions; this can be interpreted as the fact that population aging reduces CO_2_ emissions.

Other studies argued that population aging leads to an increase in pollution emissions [18,19,20]. In [18] studied the impact of population aging on SO_2_ utilizing panel data of 25 OECD countries from 1970 to 2000. It used the data in five-year steps to mitigate the nonstationary data problem. The researchers found that the young population decreases pollution emissions, while the older population increases pollution emissions. In [19] augmented standard macroeconomic emission regression by including age structure variables and cohort composition to analyze the life-cycle impact on CO_2_ emissions. They employed panel data for 26 OECD countries, for the period of 1960–2005. Using the population data in five-year steps, they mitigated the nonstationarity problem of the time-series. They concluded that the young age (people aged 15–29) and old age (people aged 60–74 and 75+) groups increase CO_2_ emissions. In [20] also focused on the role of elderly population on CO_2_ emissions from road transportation. The study used a first-order difference equation to alleviate the spurious regression problem. However, as opposed to [4], the research included the squared term of the share of elderly population to evaluate the nonlinear effect. Based on the results of 25 OECD countries from 1978 to 2008, there was an inverted U-shaped relationship between CO_2_ emissions and population aging. An increase in elderly population gradually increased CO_2_ emissions; however, beyond a 16% share of population the effect became adverse.

Furthermore, aging population may have an indirect effect on CO_2_ emissions by hindering output growth and slowing economic activities. In [21] used a panel VAR (Vector Auto Regressive) model for 21 OECD countries to estimate the effect of demographic structure change. They showed that population aging and low fertility will reduce output growth, investment and real interest rates. In [22] analyzed East Asia countries and showed that population aging significantly lowers saving rates, and it leads to worsened current account balances. In [23] surveyed empirical works of age-productivity relationship. It was evident that individuals’ job performance decreases as they become old.

[24] used a VARX (Vector Auto Regressive with Exogenous variable) model in order to estimate the effect of population aging on macro-economic variables. Using the Korean annual data from 1987 to 2017, they concluded that aging and low fertility have direct and indirect adverse effect on economic growth. First of all, aging reduces labor supply, which lowers potential growth level. Second, aging and low fertility reduce total factor productivity. Third, aging increases inflation due to reduced total supply. Last but not least, aging and low fertility rate lower investment. Based on the studies mentioned so far, it is well known that aging leads to lower economic growth, which reduces CO_2_ emissions.

Overall, the results of the preceding studies differ depending on the target countries and types of GHGs. In particular, there was no study focusing on the effects of the demographic factors in Korean regions. Since Korea’s aging trend is faster than any other, it is a good case study to analyze demographic effects on Korea’s regional CO_2_ emissions.

## 3. Regional CO_2_ Emissions and IPAT Model

### 3.1. Regional CO_2_ Emissions

In this section, we estimate regional CO_2_ emissions stemming from fossil fuels. Official Korean CO_2_ emissions data is provided by the Greenhouse Gas Inventory and Research (GIR) Center in Korea and the World Bank (WB). GIR and WB provide national-level data from 1990 and 1960, respectively. Both datasets are published annually. The recently heightened attention on energy and climate change policies warrants more comprehensive and sophisticated analyses. However, the Korean time-series data is limited to about 30 observations, while the WB’s data is limited to about 60 observations. Moreover, Korea experienced an economic growth miracle, which transformed it from the least developed country in the 1960 s into a developed country in 2000; hence, earlier observations may be not useful.

Against this context, we regard the estimation and use of regional CO_2_ emissions for comprehensive policy analysis to be useful and reliable. However, we lack official regional CO_2_ emissions data, which could be a barrier to more comprehensive analysis. Hence, as we mentioned before, we generate Korean regional CO_2_ emissions data from the energy usage. Given the carbon emission factors (Table 2) provided by the Korea Energy Economics Institute and Korea Energy Agency, as well as the formula based on the IPCC guidelines, we estimate the values in Equation (2):CO_2_ Emission = Energy Consumption × Emission Factor × 44/12(2)

While we have Korean regional energy consumption data from 1990, we only estimate CO_2_ emissions from 1998. There are two main reasons for doing so. First, Korean provinces have been standardized into 16 regions (Seoul, Busan, Daegu, Incheon, Gwangju, Daejeon, Ulsan, Gyeonggi, Gangwon, Chungbuk, Chungnam, Jeonbuk, Jeonnam, Gyeongbuk, Gyeongnam, and Jeju) since 1998. Second, in 1997, Korea experienced the Asian Financial Crisis and liberalization of the oil market. Oil had been the primary energy source until these two major exogenous shocks caused an economic structural change.

Figure 1 shows the estimated (blue dashed line) and actual values (orange solid line) of CO_2_ at the national level since we lack actual CO_2_ values on the regional level. The horizontal axis refers to the years from 1998 to 2016, and the vertical axis represents greenhouse gas emissions, which are measured in millions of CO_2_ tons. CO_2_ eq refers to the amount of greenhouse gases converted into 204 units of CO_2_.

Since we lack actual CO_2_ values at the regional level, there is a slight difference between the two lines. The mean absolute percentage error (MAPE) from the actual value is only 3.7%, which is under 5%. We deem the estimates to be sufficiently accurate for econometric analysis; thus, we proceed with the empirical analysis.

### 3.2. IPAT Model

To develop the empirical model, we adapt the IPAT model following [18,25]. The IPAT model is a simple identity function that decomposes national emissions into socio-economic factors. It is a general method of decomposing environmental impact (I) into socio-economic variables: population (*P*), affluence (A), and technology (T). In [18] applied the IPAT model to decompose SO_2_ emissions; in this study, we apply the IPAT model to decompose CO_2_ emissions.

Derivation of the empirical regression model begins with the following IPAT identity:(3)$$C_{it}=P_{it}\frac{Y_{it}}{P_{it}}\frac{E_{it}}{Y_{it}}\frac{C_{it}}{E_{it}}$$
where $C_{it}$ denotes total carbon dioxide emissions in region *i* and year *t*, *p* denotes the total population size, *Y* represents the aggregate income, and *E* is the national energy consumption. The above identity can be written more precisely in per capita terms:(4)$$C_{it}=P_{it}\times y_{it}\times e_{it}\times c_{it}$$
where $y_{it}$ is per capita income, $e_{it}$ is energy consumption per income, and $c_{it}$ is the emission factor. According to previous literature, e and *c* partly depend on the population size, per capita income, and other exogenous variables ($X_{it}$). Among exogenous variables, population age distribution plays an important role based on the life-cycle hypothesis and empirical facts [18].

From Equation (4), population age distribution has an effect on environmental quality through energy intensity ($e_{it}$) and emission factor ($c_{it}$). As we explained through previous chapters, empirical facts based on a large previous study indicate that population aging can either increase or decrease energy use. Old people prefer staying home longer than younger people due to physical limitations. They generally demand more energy for indoor activities, heating and cooling, which leads to an increase in CO_2_ emissions.

However, at the same time, old people also could decrease CO_2_ emissions since they use energy appliances in less intensive way. In terms of environmental quality, there is a consensus on a positive relationship with population aging. As life expectancy increases, old people tend to care more about environmental quality and they actively take part in politics to support favorable environmental policies.

Against these backdrops, Equation (4) may be rewritten as:(5)$$C_{it}=P_{it}\times y_{it}\times e_{it}\left(P_{it},y_{it},X_{it}\right)\times c_{it}\left(P_{it},y_{it},X_{it}\right)$$

Which reduces to:(6)$$C_{it}=f\left(P_{it},y_{it},X_{it}\right)$$

## 4. Empirical Analysis

In this section, we set up the empirical model using Korean regional panel data for 16 provinces from 1998 to 2016. We include the cubic term of the income variable and age structure variables to estimate the IPAT augmented EKC model. By extending the model, we can verify the N-shaped EKC of CO_2_ emissions and assess the role of age structure, particularly population aging, on CO_2_ emissions.

In terms of the empirical strategy, we apply the panel cointegration method since our data set consists of a relatively small cross-section and long time-series. This type of macro and regional panel data has strong time-series properties. However, many earlier and even recent studies do not deal with the time-series issue, thereby raising the problem of unreliable statistical inference [26]. To overcome this problem, we apply the FMOLS approach of [27,28] to produce asymptotically unbiased estimates and standard normal distributions that are free of nuisance parameters. FMOLS is also designed to correct residual autocorrelation and endogeneity.

### 4.1. Base Model

Based on Equation (6), our baseline estimation model includes total population (*P*), per capita income (*y*), and other exogenous variables (X), including socio-demographic factors. The baseline model takes the following form:(7)$$C_{it}=\alpha_{i}+\beta_{1}y_{it}+\beta_{2}y_{it}^{2}+\beta_{3}y_{it}^{3}+\beta_{4}OP_{it}+\beta_{5}P_{it}+\beta_{6}MAF_{it}+\beta_{7}SVC_{it}\;+\beta_{8}YOU_{it}+\beta_{9}OLD_{it}+u_{it}$$
where $y_{it}$ is per capita income, $OP_{it}$ is the oil price, $P_{it}$ is total population, $MAF_{it}$ is the ratio of manufacturing value-added to agriculture, $SVC_{it}$ is the ratio of commercial value-added to agriculture, $YOU_{it}$ is youth population, $OLD_{it}$ is old population and $u_{it}$ is the error term for region *i* at time *t*.

Equation (7) is represented in logarithmic form. We include a third-order polynomial of real per capita income at 2010 prices in order to explore the N-shaped EKC form. We include *MAF* and *SVC* as exogenous variables to incorporate the industrial structure of regions. Moreover, we add the oil price as the overall proxy of energy prices.

In terms of age distribution, we include both *YOU* and *OLD* variables to validate the robustness of the estimation. *A0014* refers to the proportion of the total population that comprises youth aged between 0 to 14 years. *P0014* is the youth dependency ratio, i.e., the ratio of the youth population to the working population.

For the *OLD* variable, *O65* is the proportion of the population aged 65 and over in the total population. *P65* is the dependency ratio of the old population, i.e., the ratio of the population aged 65 and over to the working population. Except for the dependent variable, CO_2_ emissions, all variables are taken from the Korean Statistical Information Service (KOSIS).

In summary, we construct a balanced panel data set comprising 304 observations from 16 regions for the period from 1998 to 2016. Table 3 displays the average descriptive statistics of all variables for the entire period. Per capita CO_2_ emissions have a large standard deviation between regions and are skewed to the right. In addition, the number of people over 65 years old was higher than that of 14-year-olds.

### 4.2. Panel Unit Root Test

Before estimating Equation (7), we perform panel unit root tests to verify the nonstationarity of variables. The three tests of LLC (Levin-Lin-Chu), IPS (Im-Pesaran-Shin), and MW (Maddala-Wu) [29,30,31] are most widely used for panel unit root tests in empirical works. However, these “first generation” unit root tests assume that the individual time series in the panel are cross-sectionally independent. While this may be reasonably applied in the case of micro panel data with large N, it is a rather strong assumption in the case of macro and regional panel data.

To get rid of any cross-sectional dependency, [32] develops new test statistics by including cross-section mean and its lagged value in cross-sectionally augmented Dickey–Fuller (CADF) regression with linear trends as follows:(8)$$∆y_{it}=\alpha_{i0}+\alpha_{i1}t+b_{i}y_{it-1}+c_{i}{\bar{y}}_{t-1}+d_{i0}∆{\bar{y}}_{t}+u_{it}$$
where ${\bar{y}}_{t}$ is cross-section mean of $y_{it}$, $∆{\bar{y}}_{t}$ is the cross-section mean of $∆y_{it}$ and $u_{it}$ is the error term. CIPS (Cross-sectionally augmented IPS) test statistic is constructed as follows by taking averages of the t-value of coefficient $b_{i}$ in the above CADF regression, where N and T refer to the cross-section and time series dimensions, respectively.
(9)$$\mathrm{CIPS}(N,\;T)=\bar{t}=N^{-1}\sum_{i=1}^{N}t_{i}\left(N,T\right)$$

[32] show that the limit distribution of the CIPS exists and is free of nuisance parameters. Additionally, the test has satisfactory power regardless of sample size and linear trends.

In our regional panel data set, the individual time series can also be somewhat correlated; thus, we use the CIPS test to account for any cross-sectional dependence. Table 4 shows the result of the CIPS statistics. We perform the test including and excluding the linear trend in the specification based on Equation (8), for robustness. Based on the results, we cannot reject the null hypothesis of a unit root in both specifications for any variable. For CO_2_ emissions (*lnC*) and per capita income (*lny*), we can reject the null hypothesis in one of the tests but there is no concrete evidence of stationarity. Therefore, we proceed to the cointegration test.

### 4.3. Panel Cointegration Test

Verifying the existence of unit roots from Section 4.2, we perform the panel cointegration test following [33,34,35] to detect the long-run equilibrium relationship between variables.

Table 5 and Table 6 show the results of the panel cointegration tests. We can reject the null hypothesis of no cointegration based on all the statistics of the Pedroni test, which provides strong evidence of a long-run relationship between the variables. All statistics were lower than 0, which strongly rejects the null hypothesis of 1. Among eight statistics, six statistics reject the null hypothesis of no cointegration at a 5% significance level. Hence, we can reasonably conclude that there is a long-run relationship between the variables. Furthermore, the Kao cointegration test also rejects the null hypothesis at a 10% significance level.

### 4.4. Panel Fully Modified OLS

In the empirical analysis with panel data, the nonstationarity of time-series is an important issue for the correction of statistical inference. To account for the nonstationarity property, many empirical works transform level variables into first-differenced variables [20,36,37]. The estimated coefficient in the differenced regression model represents a short-run effect. However, the EKC hypothesis and the population–environment interaction are the long-run phenomena; thus, these estimated results are not suitable for the purpose [25]. In order to conduct a correct statistical analysis and to account for long-run effects, we estimate Equation (7) by applying panel FMOLS of [27,28].

We built six models to estimate Equation (7). Models (1) and (2) are the basic models. We add a cubic term of per capita income ($y^{3}$) to reflect the EKC theory. Population (*P*) is derived from the IPAT model. Additionally, we include oil price (*OP*), manufacturing and commercial value-added ratio (*MAF, SVC*) as exogenous variables. The oil price represents the overall energy price of the regions. Manufacturing and commercial ratio represent industrial structure. To incorporate age distribution, Model (1) uses the proportion of the youth and the elderly to the total population (*A0014, O65*). However, Model (2) uses the dependency rate of the youth and the elderly (*P0014, P65*) for robustness.

Models (3) and (4) are similar to Models (1) and (2) except for the cubic term of per capita income. Models (3) and (4) represent a U-shaped or inverted U-shaped carbon–income relationship. Models (5) and (6) are estimated to check the robustness. Model (5) excludes the commercial value-added ratio, which is not significant in Table 7. Furthermore, Model (6) excludes all industrial structure variables (*MAF* and *SVC*) for robustness.

All estimation results are summarized in Table 7. The estimates can be interpreted in terms of elasticity because all variables are in logarithm form. Most estimates are highly statistically significant. In Models (1), (2), (5) and (6), the estimates of *y* and $y^{3}$ are positive, while that of $y^{2}$ is negative; thus, we can expect an N-shaped curve of the carbon–income relationship in our sample. However, we find that the roots are imaginary. Thus, we cannot support inverted U-shaped or N-shaped curves. This may be mathematically interpreted as a monotonous increase in the carbon–income relationship.

Additionally, we run the regression models without the cubic term of per capita income to find the turning point, as indicated by Models (3) and (4) in Table 7. The estimates of *y* and $y^{2}$ are negative and positive, which is indicative of a U-shaped curve. We identify the turning point incomes to be 10.0~10.3 million KRW (USD 8686~8886 in 2010) and they are found to be within our sample.

Next, we compare our results with two previous articles investigating Korean EKC. In [15] verified the U-shaped EKC, which is consistent with our results. However, the researchers identified the turning point incomes to be much higher (at the level of USD 26,400~30,000). In [16] investigated the EKC theory of CO_2_ emissions in Korea using time-series analysis and found the turning points of USD 10,119 and USD 11,711 that triggered an inverted U-shaped curve. These turning point incomes exceed ours but are closer than those of [15]. The two articles are references to our paper; however, they are not directly comparable due to differences in pollution emissions, methodology, and sample range.

Besides the income variables, the impacts of other exogenous variables are largely qualitatively consistent across Models (1) to (6). Oil price (OP), the proxy for general energy prices, exerts an adverse effect on CO_2_ emissions in Models (1) and (2). An increase in energy price leads to a decrease in fossil fuel consumption, thereby directly leading to a decrease in pollution emissions. Nonetheless, the estimated elasticity is rather small (at the level of −0.03~−0.04%). Moreover, the estimated elasticities lose their statistical significance in other models.

The estimated coefficient of the total population (*P*) is positive and significant, which is aligned with our expectations. The growing population raises the total energy consumption, thereby increasing environmental pollution. The results imply that a 1% increase in total population leads to an average of 0.9% increase in CO_2_ emissions. In the case of industrial structure, a higher manufacturing ratio seems to increase CO_2_ emission significantly. A 1% increase in manufacturing ratio leads to a 0.2%~0.3% increase in emissions. However, the effect of the commercial ratio is not statistically significant.

We now turn to the main goal of our analysis, which is to investigate the role of age structure on pollution emissions. The overall suggestion from Models (1) to (6) is that an increase in the elderly reduces CO_2_ emissions. In Models (1) and (3), we use the proportion of old people aged 65 and over in the total population and obtain estimated elasticities of −0.41% and −0.23%, respectively.

For a check on robustness, we use the old dependency ratio in Models (2) and (4) and estimate corresponding elasticities of −0.39% and −0.27%, respectively. Additionally, excluding industrial structure variables does not affect the estimates of the old dependency ratio in Model (6). The values differ slightly from each other but have the same effect qualitatively. These results are supported by previously mentioned literature [4,5,17].

There are four possible explanations for these results. First, elderly people place higher value on environmental quality; this should lead to a positive correlation between population aging and air quality [7,8].

Second, since the elderly are less active than the young, an increase in the share of the older population can lead to a decrease in pollution. Old people tend to stay home longer than the young and this may decrease transportation demand [4]. In [20] showed that elderly people account for less CO_2_ emissions in the transportation sector.

Third, even though they tend to stay home longer, their energy use intensity may be lower than that of young people, thereby leading to a potential decrease in energy consumption [3,6].

Fourth, aging may have an indirect effect on CO_2_ through hindering economic growth. From previous studies, it is well known that aging reduces output growth, investment, and productivity. As economic activity is strongly correlated with emissions, aging may have reduced CO_2_ emissions indirectly [17,21,23,24].

On the other hand, we find that the young population has a positive effect on CO_2_ emissions. Based on Models (1) and (3), a 1% increase in the proportion of young population results in a 0.22% and 0.42% increase in CO_2_ emissions respectively. The impact remains the same when using the youth dependency ratio in Models (2) and (4), where a 1% increase in youth dependency ratio leads to a 0.25% increase in emissions.

A possible explanation could be related to the “Nintendo-effect” [3] whereby children usually watch more television, use personal computers, and are heavy game users. This may lead to an increase in electricity demand and hence, more CO_2_ emissions.

Another explanation is related to household composition. Children and young population are raised by younger households, which are of have relatively larger size than aged households. Younger households are more active and drive more than other age groups; as a result, they emit more CO_2_ emissions [4,5].

### 4.5. Time Series Fully Modified OLS

From the previous sections, we could verify that the age structure plays an important role in CO_2_ emissions on a regional level. However, one could criticize that regional analysis does not account for the CO_2_ emissions from transportation for commuting and fossil-fueled power generation. For example, many people in Seoul commute to Gyeonggi province and vice versa. Thus, they emit CO_2_ or other pollutants in at least two regions. Furthermore, Korea has a central power grid system and coal-powered generations, fossil fuel containing the most carbon, are mostly located in Gyeonggi, Chungnam and Gyeongnam. The distribution of present facilities does not depend on population age structure.

Against these backgrounds, we additionally estimate the nation-wide time-series regression model with two sector-specific dependent variables, CO_2_ emissions from the transportation sector and from the residential sector, for a robustness check. By doing this, we could embrace the aforementioned limitations.

Figure 2 depicts CO_2_ emissions from the transportation and residential sectors. In Figure 2, we may notice that CO_2_ emissions from the transportation sector (blue dashed line) are continuously increasing, but these from the residential sector (orange solid line) are decreasing.

For the nation-wide time series regression, we run Equation (6) using time series FMOLS [38]. CO_2_ emissions from the transportation and residential sectors are taken from GIR and other variables are taken from KOSIS. Data span is from 1990 to 2016 based on sector-specific CO_2_ emissions.

Table 8 displays the results of time series FMOLS estimation where RES (1), RES (2) and RES (3) are results of FMOLS with residential sector CO_2_ emissions and TR (4), TR (5) and TR (6) are results with transportation sector CO_2_ emissions. We can verify that the effects of age structure variables, *A0014*, *O65*, *P0014*, and *P65* on sector-specific CO_2_ emissions are qualitatively the same as previously estimated panel models. The estimated values are somewhat larger than those of the panel models since the data is nationally aggregated and the size of the sample is much smaller.

The nonlinear relationship between CO_2_ emissions and per capita income found to be different from the sectors. The results of RES (1) to RES (3) indicate that there is a U-shaped curve between the two variables and the estimated turning point income is about USD 11,000. This indicates that there is a strong positive relationship between income and residential CO_2_ emissions. On the other hand, the results of TR (1) to TR (3) show the opposite relationship. We can find an inverted U-shaped curve between the two variables and the estimated turning point income is about USD 30,000.

## 5. Discussion

In this study, we explore the impact of age structure on CO_2_ emissions in Korea. We combine the IPAT model with standard EKC regression and additionally include population aging and youth variables. Using the regional panel data of 16 provinces from 1998 to 2016, we conduct the panel unit root and cointegration tests. We verify the fact that there is a long-run relationship between variables. In order to correct for nonstationarity, we employ the panel FMOLS.

The main empirical results are as follows. First, in the regression with the third-order polynomial of per capita income, we establish the N-shaped curve between CO_2_ emissions and income. However, we are unable to mathematically derive the real-valued turning point incomes. In additional regression, we verify the U-shaped relationship between CO_2_ emissions and per capita income with a turning point income of approximately USD 8,800. Hence, we conclude that the classical inverted U-shaped curve does not exist in our empirical analysis and income monotonically increases CO_2_ emissions.

Second, using two types of population aging variables, namely the proportion of the total population aged 65 and over and the old dependency ratio, we conclude that population aging reduces CO_2_ emissions. Our results are supported by several previous studies focusing on behavioral perspectives that confirm the fact that although old people spend significantly more time at home, they are relatively less active than other age groups. Furthermore, as the elder people live in smaller houses and have significant preferences in terms of air quality, aging population may decrease CO_2_ emissions. Besides, the relationship between population aging and CO_2_ emissions can be interpreted by economic channel. Based on the studies we considered, aging slows down economic activities due to reduced labor supply. It additionally lowers productivity, generates inflation, and lowers investment rates.

Third, the young population increases CO_2_ emissions. We test two types of variables, namely the proportion of population aged under 14 years and the youth dependency ratio. With a 1% increase in the proportion of the youth results in a 0.2% increase in CO_2_ emissions. The result is supported by the “Nintendo-effect,” whereby children watch more television and use electrical appliances more intensively. Moreover, younger households with children are more active than other households.

## 6. Conclusions

These empirical results suggest important policy agendas. Korea has been confronting two problematic issues: rising CO_2_ emissions and the aging population. The analysis shows that regional CO_2_ emissions tended to continuously rise with income growth. In other words, the EKC theory that greenhouse gas emissions would decrease beyond a certain level of income is hardly proved in relation to CO_2_ emissions in Korea. Based on our empirical results, we cautiously expect that it is not easy to reduce greenhouse gas emissions with the existing economic growth policy.

However, due to the different behavioral patterns from the aging population and the youth, CO_2_ emissions can be reduced. We verify that old people emit less carbon emissions, whereas the youth emit more. As population aging and low fertility deepen, we expect that the effects of population aging on emissions will dominate those of the youth. Therefore, policymakers should consider the additional effect of population age structure on long-term greenhouse gas emissions.

Korea is also facing a low fertility problem. Various policies are implemented to raise the birthrate for economic vitality. However, according to this study, these policies may increase carbon emissions in the long-run. In order to resolve demographic and carbon emission problems comprehensively in Korea, policymakers should focus on reducing the carbon contents of products to offset the emissions increased by young people.

## Figures and Tables

**Figure 1 ijerph-17-02972-f001:**
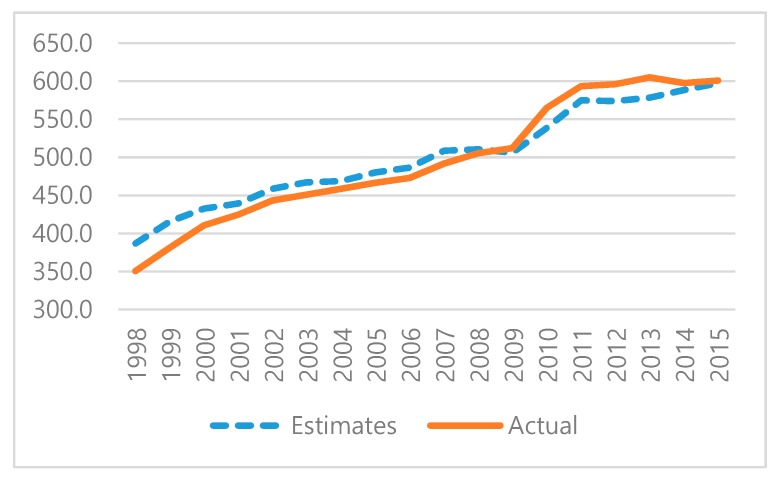
CO_2_ Emissions in Korea (mil. ton of CO_2_ eq).

**Figure 2 ijerph-17-02972-f002:**
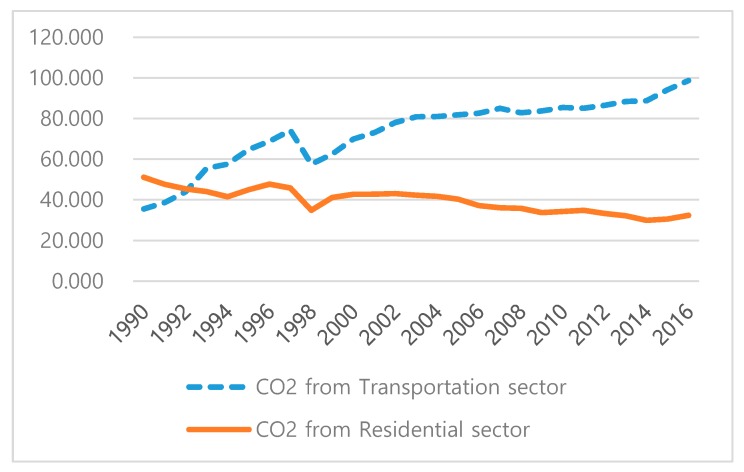
CO_2_ Emissions from transportation and residential sector in Korea (mil. ton of CO_2_ eq).

**Table 1 ijerph-17-02972-t001:** Literature survey on the effect of aging and low fertility on environmental quality.

Effect	Literatures	Effect of Aging and Low Fertility on Environmental Quality
Direct effect	[4] Liddle (2011)	Aging reduces CO_2_ emissions
	[5] Liddle and Lung (2010)	Aging reduces CO_2_ emissions
	[18] Menz and Kuhling (2011)	Young-age reduce SO_2_ & Old-age increase SO_2_
	[19] Menz and Welsch (2012)	Young and old-age increase CO_2_
	[20] Okada (2012)	Inverted U-shape between age distribution and CO_2_
Indirect effect	[17] Dalton et al. (2008)	Aging reduces labor supply
	[21] Aksoy et al. (2019)	Aging and low fertility reduce output growth, investment and real interest rates
	[22] Kim and Lee (2007)	Aging lowers saving rates
	[23] Shirbekk (2004)	Aging reduces labor productivity
	[24] Jo et al. (2019)	Aging reduces labor supply, productivity, and investment & Aging increases inflation rate

**Table 2 ijerph-17-02972-t002:** Carbon emission factor.

Emission Factor (EF)	Coal	Petroleum	Electricity	Urban Gas
C Ton/TOE	1.059	0.829	0.553	0.637

**Table 3 ijerph-17-02972-t003:** Descriptive statistics.

Variable	Mean	Standard Deviation	Skewness	Kurtosis	Obs
*C*	31.67	27.93	1.31	4.20	304
*y*	23.60	10.64	1.80	6.11	304
*OP*	96.01	22.98	0.15	1.78	304
*P*	3,049,092	2,924,078	2.04	5.81	304
*MAF*	34.21	9.67	0.12	1.96	304
*SVC*	95.63	28.94	0.35	2.30	304
*A0014*	18.06	3.18	0.09	2.22	304
*P0014*	25.43	4.81	0.02	2.09	304
*O65*	10.69	3.84	0.38	2.46	304
*P65*	15.18	5.93	0.56	2.73	304

**Table 4 ijerph-17-02972-t004:** Cross-sectionally augmented panel unit root (CIPS) test results.

Variable	No Trend	Trend
*lnC*	−2.52	−2.82 ***
*lny*	−2.69 *	−1.93
*lnOP*	−1.75	−1.65
*lnP*	−1.06	−1.40
*lnMAF*	2.60	1.70
*lnSVC*	2.61	1.70
*lnP0014*	−2.34	−1.46
*lnP65*	−2.20	−0.61
*lnA0014*	−2.35	−2.06
*lnO65*	−2.19	−0.98

***: *p*-value < 0.01, *: *p*-value < 0.10.

**Table 5 ijerph-17-02972-t005:** Pedroni cointegration test results.

	No Trend Statistic	Trend Statistic
**Panel PP**	−1.54 *	−2.80 ***
**Panel ADF**	−1.47 *	−3.01 ***
**Group PP**	−2.27 **	−3.36 ***
**Group ADF**	−2.17 **	−3.51 ***

***: *p*-value < 0.01, **: *p*-value < 0.05, *: *p*-value < 0.10; PP stands for Phillips-Peron, and ADF stands for Augmented Dickey-Fuller statistics; Panel PP and ADF test for homogenous alternative for all i, and Group PP and ADF test for heterogeneous alternative for all i.

**Table 6 ijerph-17-02972-t006:** Kao cointegration test results.

	MDF	DF	ADF	UMDF	UDF
**Statistics**	−1.60 *	−1.59 *	−0.89	−1.61 *	−1.59 *

*: *p*-value < 0.10.

**Table 7 ijerph-17-02972-t007:** Panel fully modified ordinary least squares (FMOLS) estimation results.

Var	Model (1)	Model (2)	Model (3)	Model (4)	Model (5)	Model (6)
**y**	12.84 ***	12.61 ***	−2.05 ***	−2.03 ***	12.83 ***	13.37 ***
**y^2^**	−4.11 ***	−4.04 ***	0.44 ***	0.44 ***	−4.11 ***	−4.23 ***
**y^3^**	0.46 ***	0.45 ***			0.46 ***	0.47 ***
**OP**	−0.03 **	−0.04 **	−0.03	−0.03	−0.02	−0.02
***P***	0.94 ***	0.92 ***	1.01 ***	1.01 ***	0.94 ***	0.91 ***
**MAF**	0.23***	0.23 ***	0.32 ***	0.31 ***	0.19 ***	
**SVC**	−0.07	−0.07	−0.10	−0.10		
**A0014**	0.22 ***		0.42 ***		0.24 ***	
**O65**	−0.41 ***		−0.23 ***		−0.42 ***	
**P0014**		0.25 ***		0.34 ***		0.13 **
**P65**		−0.39 ***		−0.27 ***		−0.37 ***
**Adj-R^2^**	0.99	0.99	0.99	0.99	0.99	0.99
**Obs**	288	288	288	288	288	288
**TP (USD)**			$8886	$8686		

***: *p*-value < 0.01, **: *p*-value < 0.05.

**Table 8 ijerph-17-02972-t008:** Time series FMOLS estimation results.

Var	RES (1)	RES (2)	RES (3)	TR (4)	TR (5)	TR (6)
**y**	−24.24 ***	−11.93 ***	−12.08 ***	7.79 ***	6.95 ***	8.45 ***
**y^2^**	4.58 ***	2.36 ***	2.41 ***	−1.09 **	−0.98 **	−1.22 ***
**OP**	−0.15 ***	−0.08 ***	−0.04 ***	−0.28 ***	−0.28 ***	−0.34 ***
***P***	25.92 ***	11.50 ***	12.82 ***	−6.79	−6.69	−0.60 ***
**MAF**						0.07 ***
**SVC**			−0.37 ***			
**A0014**	5.46 ***			0.69		−0.96 ***
**O65**	−3.49 ***			−1.96 **		−0.72 ***
**P0014**		3.26 ***	3.57 ***		1.38	
**P65**		−1.98 ***	−1.93 ***		−2.25 **	
**Adj-R^2^**	0.64	0.87	0.90	0.96	0.97	0.96
**Obs**	26	26	26	26	26	26
**TP (USD)**	$12,198	$10,832	$10,603	$30,826	$29,990	$27,607

***: *p*-value < 0.01, **: *p*-value < 0.05.

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
