# Peer review of "Do Aging and Low Fertility Reduce Carbon Emissions in Korea? Evidence from IPAT Augmented EKC Analysis"

_ijerph, 2020, doi:10.3390/ijerph17082972_

Round 1

Reviewer 1 Report

This paper selects population as driving factor to study its impact on South Korea's carbon emissions. In addition to the number of population, it also studies and analyzes the impact of population age on carbon emissions. With the aggravation of the aging population in South Korea, such kinds of research and analysis has certain practical implications for policy makers to formulate long-term energy strategy.

1. please mark the full name when IPAT model appears for the first time;   

2. on row 217, eit is not per capita energy consumption, but energy consumption per unit capital or income;    

3. please clarify all the variable description in equation 7;

4. In the last part of the results and conclusions of this paper, in addition to using empirical data to clarify the rules, we can also use such rules for further analysis, including predicting the impact of future population aging changes on carbon emissions, stimulating fertility while also increasing carbon emissions, at the same time, further reducing the carbon content of products, which can offset the emissions of young people increase.

Reviewer 2 Report

“aging” and “ageing” are mixed. Please unify to either.

l. 107 & 109
CO => CO_2 ?

l. 85
The definition of each symbol is ambiguous. What do i and t in “E_it” and “Y_it” represent? Or it is easier to read what is described later.
What does “\epsilon” represent?
Also, is there no subscript for "\alpha"?

l. 164
Here, “Table 1” appears, but this table is not cited in the text. Cite this table early in Section 2 / Paragraph and follow along with it to make the comparison easier for the reader. Also, please write the reference number by the authors in the table.

l. 193
The author would write in the figure what the horizontal and vertical axes represent in "Figure 1". The author should also specify what “mil. Ton of CO2 eq” represents.

l. 252
It is not clear what “SD / SK / KT” in “Table 3” represents. It is also unclear how the value is evaluated as data. (Are these data proper to analysis?)
Also, “C / P / PD” has a larger SD value than the Mean value. Do these have any effect on the analysis?

l. 262
There is no explanation of what “u_it” is.

l. 265
There is no explanation of what "N" is.

l. 285
There is no description of “PP” or “ADF”. It is also necessary to explain that all the numbers in “Table 5” are negative.

l. 297, l. 305
Please explain the details of “Model (1)-(4)”. Also, it is confusing with the notation such as equation (1),(2),.. Please replace to different notation. Also, please explain what the numbers in “Table 7” indicate. In addition, is there any notation of how different "\alpha_i" is?

l. 321-322
If you create a model that includes strongly correlated variables such as “total population” and “population density” at the same time, you may get strange results in which one takes a positive value and one takes a negative value. In that case, you may need to create a model that contains only one of the variables and check the results.

l. 332 etc.
The word “negative” makes the reader confused, so change to another word. (The word “negative” also has the meaning of something bad, so the word in the sentence will remain unclear to the reader whether CO2 emissions will increase or decrease)

l. 339
It is also necessary to consider the ratio of youth and elderly, but how about the industry in each region? Modern industrial and urban areas are likely to emit more CO2 and have a larger population of prime young people. Conversely, rural areas and areas with a large population of retirees will emit less CO2 and have a larger population of older people. (It did not reflect the behavior of the inhabitants but may have been the way in which the inhabitants were divided to match the characteristics of the region. Such kind of consideration of regional features should be noted in here and future work.)

l. 379
Arrange “Table 8” so that it is not divided.
Also, since the coefficients of each result are often different in sign, it is easier for the reader to compare the results if the outline of each model is known as in “Figure 1”. (Then, the discussion of l. 381-391 could be made more detailed.)

l. 473
20. Okada, A. Is an increased elderly population related to decreased CO2 emissions from road transportation
=> … CO2 emissions from road transportation?

l. 489
28. Waggoner, P. E.; Ausubel, J. H. A framework for sustainability science: A renovated IPAT identity. Proceedings of the National Academy of Sciences, 2008, 99(12), 7860-7865.

=> 2002
c.f. https://www.pnas.org/content/99/12/7860.short
